# Raman sideband cooling
# in optical tweezer arrays for Rydberg dressing

Nikolaus Lorenz[1,2]⋆, Lorenzo Festa[1,2], Lea-Marina Steinert[1,2] and Christian Gross[1,2,3]

**1** Max-Planck-Institut für Quantenoptik, 85748 Garching, Germany
**2** Munich Center for Quantum Science and Technology (MCQST), 80799 München, Germany
**3** Physikalisches Institut, Eberhard Karls Universität Tübingen, 72076 Tübingen, Germany

⋆ nikolaus.lorenz@mpq.mpg.de

## Abstract

Single neutral atoms trapped in optical tweezers and laser-coupled to Rydberg states provide a fast and flexible platform to generate configurable atomic arrays for quantum simulation. The platform is especially suited to study quantum spin systems in various geometries. However, for experiments requiring continuous trapping, inhomogeneous light shifts induced by the trapping potential and temperature broadening impose severe limitations. Here we show how Raman sideband cooling allows one to overcome those limitations, thus, preparing the stage for Rydberg dressing in tweezer arrays.



# 1 Introduction

Recently, optical tweezers gained increasing interest, as they allow for fast preparation of single atoms in one, two or three dimensions with configurable geometries [1–9]. Sizeable interactions over long distances of several micrometers can be induced by excitation to atomic states with large principal quantum number, so called Rydberg states. The full control over spatial geometries and the mutual interactions makes Rydberg atoms in optical tweezers an excellent platform for quantum information [9–20], quantum simulation [21–25] and quantum metrology [8, 26–29]. Rydberg arrays naturally feature state dependent interactions, a fact that renders them particularly suited for implementing spin models, in which many fundamental many-body problems can be studied [30–40].

In contrast to direct excitation to Rydberg states, interactions between ground states can be induced by admixing Rydberg states via near-resonant coupling. This so-called Rydberg dressing allows to engineer interactions over long distances among ground state atoms [41–44]. Coherent dressing induced interactions were realised so far in a pair of microtraps, optical lattices or bulk systems [45–48], but not in larger optical tweezer arrays. The central challenge is to overcome inhomogeneities of the trapping potentials, which are typically on the order of 10 % of the total trap depth. Away from magic trapping wavelengths, they result in trap-to-trap variations of the light shift and thus unequal laser detunings for dressing experiments. In addition, the atoms are typically at microkelvin temperatures after being loaded into the tweezers. This leads to thermal broadening of the Rydberg transition. In the weak dressing regime, where the probability for an atom to be in the Rydberg state $\beta^2 = \Omega^2/4\Delta^2$ is small, the induced interaction strength is given by $U_0 = \Omega^4/8\Delta^3$, with Rabi frequency $\Omega$ and detuning $\Delta$ [42,44,45]. The maximal reachable ratio of the interaction timescale to dissipation rate due to Rydberg state decay with rate $\Gamma_r$ is then $R = \Omega^2/2\Delta\Gamma_r$, where the Rabi frequency is limited by the available laser power. Hence, for high coherence one has to work at a comparably small detuning, typically of a few megahertz. This is in the same order of magnitude as inhomogeneities and Doppler broadening. Reducing the temperature of the atoms in the array can overcome both limitations. The Doppler shift is directly related to the temperature, while the inhomogeneities are linked to the temperature limited trap depth required to hold the atoms in place. Raman sideband cooling provides a very efficent way to optically cool the atoms to the motional ground state in tight traps [49–51]. It has been successfully applied to single or few tweezers [52–55]. Here we demonstrate Raman sideband cooling and single atom trapping in a tweezer array for $^{39}$K atoms. We cool the atoms close to the ground state and show that inhomogeneous light shifts and thermal broadening can be reduced to a negligible level. These improvements pave the way for coherent Rydberg dressing in tweezer arrays.

# 2 Experimental setup

The experiment starts with loading $^{39}$K atoms from a Zeeman slower into a magneto optical trap. To generate the optical tweezers, trapping light from a high power fibre laser operating at 1064 nm is sent to a liquid crystal light modulator [1,2,56]. The light modulator provides local control of the phase of the light in Fourier plane and is imaged onto an in-vacuum objective with a working distance of 16.75 mm and a numerical aperture of 0.6. The objective features a central hole with 8 mm diameter to offer optical access with large beam size also from the vertical direction. The central part of the beam that is sent onto the objective is blocked beforehand, to prevent unfocussed light at the atoms. Numerical simulations of the objective showed an increase of the wings of the point spread function, but no reduction of resolution due to the central hole. The fluorescence light from single atoms is collected on a scientific-CMOS

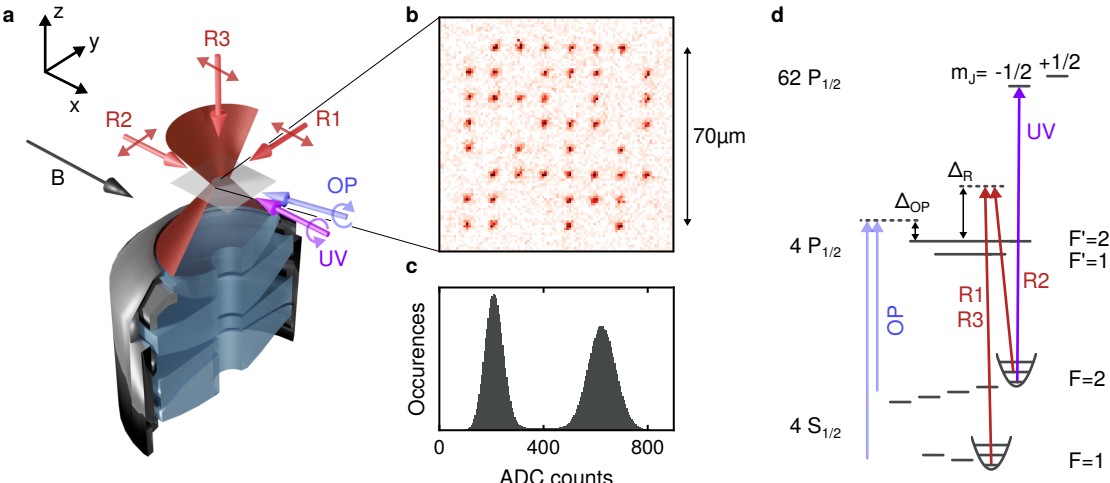

Figure 1: **a.** Sketch of the experimental setup with the high NA objective and the optical tweezers as well as the Raman beams (R1, R2, R3), the optical repumper (OP) and the UV beam for Rydberg excitation. Polarisations of the beams are indicated by arrows. **b.** A single fluorescence picture with 64 tweezers and **c.** Histogram of $20 \cdot 10^3$ experimental shots and 64 individual tweezers ($\approx 1$ million data points), demonstrating high fidelity readout. **d.** Relevant levels and transitions of $^{39}K$ for the Raman cooling and Rydberg excitation. The $4P_{3/2}$ states (D2 line) used for MOT and molasses cooling are not shown.

camera using the same objective [57]. We observe no visible degradation of the raw images due to the hole. To load and image single atoms in the tweezers we use an optical molasses on the D2 line of potassium with a detuning of $1.5\,\Gamma$ from the $F = 2$ to $F' = 3$ cycling transition, with the natural linewidth $\Gamma \approx 2\pi \cdot 6\,$MHz [58]. Both molasses and trapping light are chopped out of phase at 1.4 MHz to avoid heating from the strongly anti-trapped excited state [59,60]. We apply a parity projection pulse [61,62], to prepare single atoms with a probability of about 50%. The atoms are prepared in the $F = 2, m_F = +2$ stretched state by optical pumping on the D1 transition at 1.5 G quantisation field in the x-direction. For loading and imaging we use an average optical power of approximately 15 mW per tweezer. The chopping averaged trap depth is $906 \pm 103\,\mu$K, where the error denotes the inhomogeneity dominated standard deviation from averaging 64 individual traps. Due to optical abberations in the path of the trap light, the tweezers are not homogeneous with variations of $\pm 11.4$% in the trapping potentials. The waist in the radial direction is $0.9\,\mu$m with mean trapping frequencies $\omega_r = 158\,$kHz in the radial and $\omega_a = 25\,$kHz in the axial direction. We measure a vacuum limited lifetime of single atoms without chopping the trap of $81 \pm 8\,$s. For all experiments a first fluorescence picture is taken after loading the atoms to determine which traps are loaded in an individual run. After the respective experiment a second picture is taken to determine the atom loss.

For Raman sideband cooling and sideband spectroscopy we use three perpendicular beams to address the radial or axial trap axes, as shown in Fig. 1. The Raman beams are $\Delta_R = 40\,$GHz blue detuned from the $4P_{1/2}$ state. All Raman beams have linear polarisation, aligned along the y axis (R2) or along the quantisation axis (R1 + R3), to prevent vector light shifts [59]. The waists at the atoms are $250\,\mu$m with intensities of $1.6\,$W/cm$^2$ (R2) and $0.9\,$W/cm$^2$ (R1 + R3). We measure Rabi frequencies of $2\pi \cdot 43\,$kHz for driving the $F = 2, m_F = +2$ to $F = 1, m_F = +1$ ground state transition on the carrier. During Raman sideband cooling and sideband spectroscopy the trap is not chopped and ramped to 225% of the initial power, which increases the mean trapping frequencies by a factor of 1.5 to $\omega_r = 236\,$kHz and $\omega_a = 38\,$kHz. The laser

power is equal to the peak power during chopping and thus imposes no limit for scaling the amount of tweezers. The optical repumpers for Raman sideband cooling are the same beams used for state preparation, but now blue detuned $\Delta_{OP} = 80 \pm 30$ MHz from the in-trap $4S_{1/2}$ to $4P_{1/2}, F = 2$ resonance to prevent heating from the anti-trapped excited states [59]. The error denotes the standard deviation due to the light shift from the 64 individual tweezers. Each beam has a peak intensity of about 4 mW/cm$^2$. To excite to high-lying Rydberg states we use the direct transition from the $4S_{1/2} F = 2, m_F = 2$ ground state to the $62P_{1/2}, m_J = -1/2$ state in the ultra violet regime (UV) at 285.88 nm, as shown in Fig. 1. The UV setup is described in more detail in Appendix A.

## 3 Rydberg spectroscopy

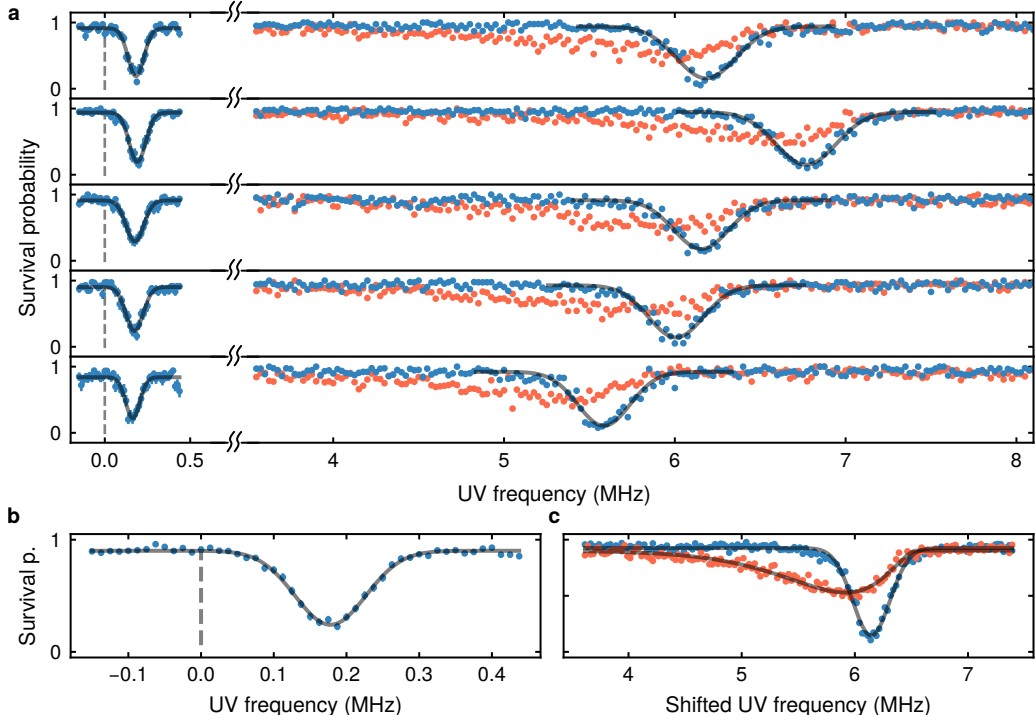

Figure 2: Effectively improving tweezer inhomogeneities by Raman sideband cooling. **a.** Spectroscopy with (blue) and without (orange) cooling at 20 percent of the tweezer power used for loading (right) and with cooling at the minimum of 5 permil power (left) for five exemplary tweezers. The grey dashed line at zero marks the free space resonance. For the cooled measurements Gaussian fits are shown in grey. **b.** Average of the individual tweezers at 5 permil trap power. **c.** The data for 20 % trap power has been shifted to a common centre of the Gaussian fit before averaging. The non-cooled data is fitted with a Maxwell Boltzmann distribution.

In order to perform dressing experiments in the far detuned regime, the difference in the light shifts of the individual traps need to be sufficiently small to ensure uniform interactions over the whole array. Typical Rabi frequencies and therefore also typical detunings for alkali tweezer experiments are in the range of a few megahertz. To achieve uniform interactions over the array, this puts a limit on the order of 100 kHz for the difference in light shifts of the individual tweezers. In order to quantify the inhomogeneities of the trapping potentials we perform spectroscopy on the UV transition.

In the first measurement we ramp the tweezer power to 20 % of the initial value and perform a Rydberg spectroscopy. This is the lowest power possible before losing the atoms, given the temperature without Raman sideband cooling. To make sure interaction effects between the atoms are small, we use a spacing of 20 μm between the atoms. To keep the power levels for the intensity stabilisation similar between different experiments, we use a 5 x 5 square array, but due to the strong focusing of the UV beam to a waist of 20 μm we only evaluate the central column of the array, which is aligned along the UV beam. The atoms are illuminated for 100 μs with a weak UV pulse and the atom loss is measured. In Fig. 2 the data for the individual tweezers with and without cooling is shown. Due to the inhomogeneous trapping potentials, the individual lines are shifted by more than 1 MHz with respect to each other. To evaluate the line shape and extract the temperature, the individual tweezer lines are shifted to a common centre determined by a Gaussian fit and averaged, as shown in Fig. 2 c. The line shape in the non-cooled measurements shows a clear asymmetry which is fitted well by a Maxwell Boltzmann distribution $f(E) = E^2 e^{-E/k_B T}/2(k_B T)^{3/2}$ with $T = 16 \pm 1.3$ μK. Note that the temperature $T$ depends on the trap depth $V_0$ since $T \propto \sqrt{V_0}$, assuming adiabaticity [63]. The same measurement is repeated with Raman sideband cooling, which results in a narrower line with a Gaussian shape (FWHM $= 386 \pm 21$ kHz) and no detectable asymmetry.

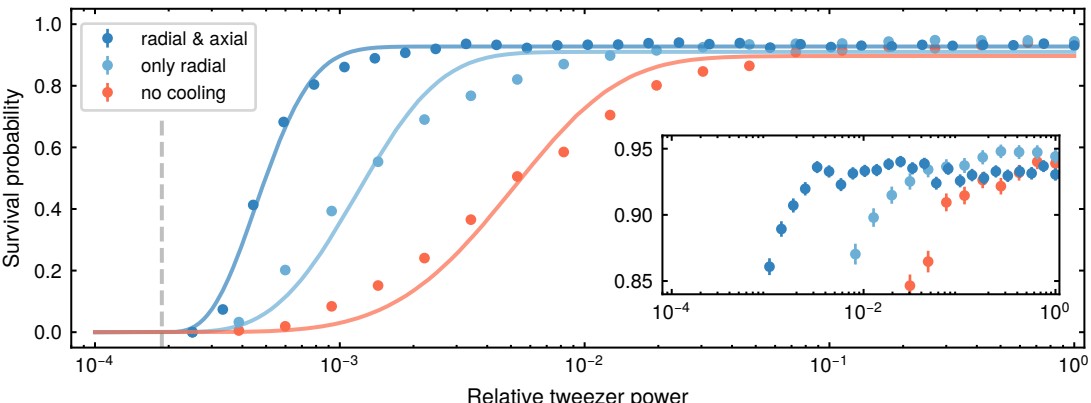

Figure 3: Atomic survival probability when adiabatically decreasing the trap depth. The data is the average of 64 tweezers with full Raman sideband cooling (dark blue), cooling only on the radial axis (light blue) and no additional cooling (orange). The solid lines are theory fits of a Boltzmann distribution, taking into account gravity. Due to gravity, for relative powers below $2 \cdot 10^{-4}$ (dashed grey line) the atoms are not confined in z anymore. The inset shows a zoom in the y axis. We attribute the mismatch of the tweezer averaged data and the fit for higher powers to tweezer-to-tweezer inhomogeneities.

The reduced temperature after cooling further allows us to ramp the trap to much lower power without atom loss. We perform the same measurement with cooling and ramp the tweezers to 5 permil of the initial power without additional atom loss. We observe a narrower line with a full width half maximum of $112 \pm 9$ kHz. At this power level, the absolute variation of the lines is about 40 kHz, smaller than the observed width of the lines and below our target inhomogeneity of 100 kHz. The measured line width is still larger than the theoretical width of the transition of $2\pi \cdot 1$ kHz, given by the blackbody reduced lifetime of $\tau_r = 160$ μs. We attribute this to the width of the laser, which we estimate from this measurement to be approximately 100 kHz.

In summary Raman sideband cooling improves thermal broadening and allows to hold atoms at very low trap depths. Ramping adiabatically, further cools the atoms and reduces the thermal

broadening even more. With cooling we are able to reduce the inhomogeneous trap shifts to the order of a few tens of kHz, while without it, we would need tweezers inhomogeneities below 1 % at minimal possible trap depth for the same absolute shift between all the traps.

# 4 Characterisation of the atomic array

As achieving the lowest possible trap depth is crucial, we now investigate the adiabatic ramp-down more closely. We use a square 8 by 8 pattern with $10\,\mu$m spacing for better statistics, shown in Fig. 1. The trap depth of the tweezers without any ramp-down is determined by measuring the light shift of the $4S_{1/2}\,F=2, m_F=2$ to $4P_{1/2}\,F=2, m_F=2$ transition. The sequence for the rampdown measurement is similar to the one discussed above. After Raman sideband cooling the traps are ramped down during 100 ms to ensure adiabaticity. The atoms are then held for 100 ms at a low power. Afterwards the traps are ramped up and the second image is taken. The results for different cooling configurations are shown in Fig. 3. If we do not apply any cooling we start losing atoms around 10 % of the initial trap depth, corresponding to a depth of about $100\,\mu$K. While only cooling the radial modes improves this limit to a few percent of the initial power, full cooling of all axes allows us to hold the atoms even at about 5 permil of the initial power (a trap depth of $3.7\,\mu$K), an improvement of two orders of magnitude. The final trap depth is indeed limited by gravity, such that even further improvement is possible by the addition of a vertical lattice [29]. For a theoretical model we follow the analysis from [63]. We first calculate the trap depth for a given power, including gravity. At a factor of $2 \cdot 10^{-4}$ of the initial power, gravity opens the trap such that the atoms are not confined anymore. To fit the temperature we assume a Maxwell Boltzmann distribution $f(E)$ and calculate the survival probability as $P_{surv}(E)=\int_0^E f(E')dE'$. Details are given in Appendix D. We extract average temperatures of $T=3.31\pm0.08\,\mu$K, $11.8\pm0.8\,\mu$K and $40.7\pm2.2\,\mu$K at the initial trap depth of $V_0=906\,\mu$K for the fully Raman cooled, only radial Raman cooled and molasses-only cooled atoms, respectively. The errors are the standard deviation from individual fits for all tweezers.

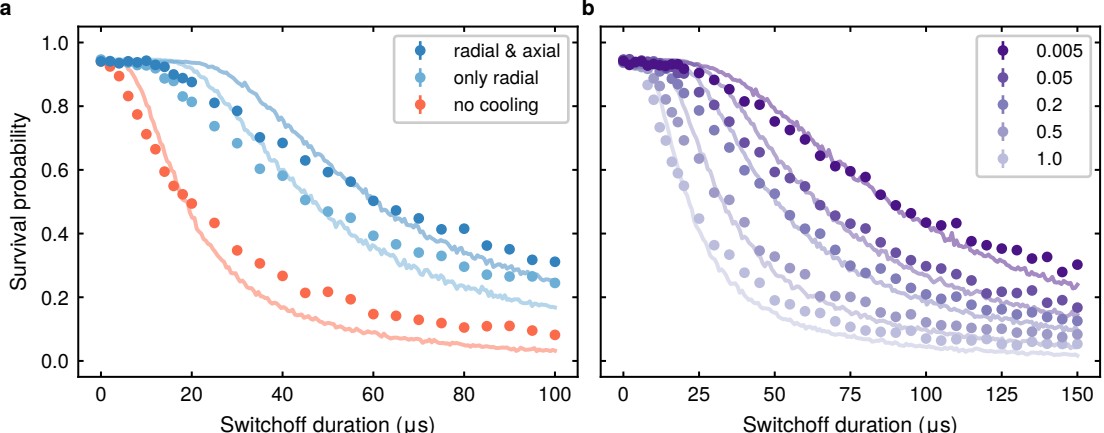

Figure 4: Survival probability for different trap off times. **a.** Comparison of atoms with Raman sideband cooling (dark blue), with only cooling the radial direction (light blue) and with no additional cooling (orange). **b.** Full cooling of radial and axial axes, but the trap power before the switch-off is varied. The label is the ramp-down factor from 1 (no rampdown) down to 0.005. The solid lines are numerical simulations of the survival probability.

Being a critical requirement for Rydberg dressing experiments in optical tweezers, Raman sideband cooling is also advantageous for experiments that rely on the trapping light to be switched off during the "physics phase" [11,12,20,22,25,47]. We now show how the survival probability after switching the trap off is improved by Raman sideband cooling. In a first measurement the trap is always ramped down to 20 % of the initial power before it is switched off (Fig. 4). Without cooling atom loss starts after $2\,\mu s$, while cooled atoms survive without additional loss up to $15\,\mu s$, an improvement by a factor of 7. We simulate the atom loss with a Monte Carlo simulation by first drawing the initial position $x$ and velocity $v$ of an atom from normal distributions with widths $\delta v = \sqrt{k_B T/m}$ and $\delta x = \sqrt{k_B T/m\omega_i^2}$. Here $T$ is the temperature, $k_B$ the Boltzmann constant, $m$ the mass of the atom and $\omega_i$ the trapping frequency in the axial or radial direction. We calculate the new position after free flight for the trap-off time and finally extract the energy of the atom instantaneously after the light field has been switched back on. The atom is considered to be lost if the final energy is larger than the trap depth. Using the temperature as a variable parameter in these simulations, we match the prediction to the data. Adiabatically lowering the trap depth $V_0$ before release cools the atoms [63]. This effect can be seen in Fig. 4, where we always apply Raman sideband cooling and scan the power to which the trap is ramped before it is switched off. We observe less atom loss and lower temperatures for lower trap depth, down to a fitted temperature of the atoms of $T = 200\,nK$ at release and recapture at 5 permil of the initial trap depth.

## 5  Rydberg Rabi oscillations

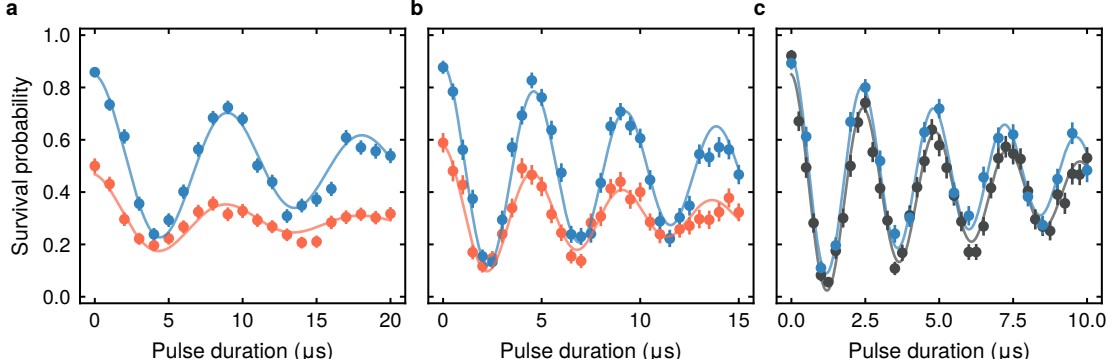

Figure 5: Resonant Rabi oscillations between the ground $F = 2, m_F = 2$ and the Rydberg $62P_{1/2}, m_J = -1/2$ states, averaged over five tweezers. Oscillations with (blue) and without (red) Raman sideband cooling and fit of an exponentially damped oscillation for Rabi frequencies of **a.** $2\pi \cdot 100\,kHz$ and **b.** $2\pi \cdot 200\,kHz$. **c.** Rabi oscillations of $2\pi \cdot 400\,kHz$ with (blue) and without (black) switching off the trap. Both measurements were performed with Raman cooling and ramping the trap to 5 permil.

Reduced thermal and inhomogeneous broadening is also advantageous for resonant Rydberg experiments and, in fact, trap switchoff is not required in a cold configuration at low trap depth. To demonstrate this we measure coherent Rabi oscillations between the ground and Rydberg state. The number of coherent oscillations $N$ is limited by several factors, such as intensity noise and beam pointing, Doppler shift as well as line width and phase noise of laser [64]. Oscillations for different Rabi frequencies are shown in Fig. 5 for different trapping configurations, where Raman sideband cooling always improves the number of coherent oscillations.

Also the amplitude of the oscillations is improved significantly in the free flight configuration, reducing state preparation and measurement errors.

The coherence improvement is due to the reduction of the Doppler shift $\Delta_D$, which is given by $\Delta_D = k \cdot v$ with the wave vector $k = 2\pi/\lambda$ and the wavelength $\lambda$ of the Rydberg excitation laser. With the temperatures extracted from the switch-off measurement, we calculate Doppler shifts of $\Delta_D = 2\pi \cdot 160\,\mathrm{kHz}$ before and $2\pi \cdot 50\,\mathrm{kHz}$ after Raman sideband cooling at a lowered trap depth of 20 %. The remaining dephasing at higher Rabi frequencies is most likely due to intensity noise and beam pointing fluctuations. The latter translate to the former given the small beam size used in the experiment. Intensity fluctuations limit the number of coherent oscillations to $N = \sqrt{2}/\pi\sigma_I$ with the standard deviation of the pulse area $\sigma_I$ [20]. Intensity fluctuations of about 7 % or beam pointing fluctutations of a few micrometers would explain the observed dephasing. At the current noise level, switching the trap off to eliminate inhomogeneous light shifts is not required anymore at the lowest trap powers enabled by Raman sideband cooling. Comparing Rabi oscillations with and without trapping light on, while adjusting the detuning for the remaining mean light shift of 200 kHz, we do not observe any notable difference in the performance (see Fig. 5).

## 6 Conclusion

In this paper we report on the first realization of potassium-39 optical tweezer arrays to study Rydberg enabled many-body spin physics. We demonstrated coherent coupling to Rydberg states with p-orbital symmetry and showed how to reduce limiting broadening of the Rydberg transition line. This was enabled by Raman sideband cooling, which allows one to overcome the severe limitations of inhomogeneous trapping potentials, common in optical tweezer arrays. We also showed that for on-resonance experiments the combination of Raman sideband and adiabatic cooling enhances the available time to study many-body physics by one order of magnitude. The improvements presented push both thermal broadening and inhomogeneous shifts into the kHz range, enabling Rydberg dressing experiments on the optical tweezer platform [16, 26, 27, 30, 31, 33, 35, 37, 65, 66].

## Acknowledgements

We thank M. Duda, A-S. Walter, S. Hirthe, J. Adema, P. Osterholz and R. Eberhard for help with the experimental setup and Z. Chen for feedback on the manuscript. We also thank A. Mayer and K. Förster for technical contributions. We further thank D. Ohl de Mello for sharing the code for the trap switch-off simulation.

**Funding information.** This project has received funding from the European Research Council (ERC) under grant agreement 678580 (RyD-QMB) and the European Union's Horizon 2020 research and innovation program under grant agreement 817482 (PASQuanS). We also acknowledge funding from Deutsche Forschungsgemeinschaft within SPP 1929 (GiRyd) and funding by the MPG.

## A UV setup

The ultraviolet light (UV) is generated by amplifying light at 1143.5 nm in a Raman fiber amplifier to 10 W and frequency doubling it in two consecutive cavity enhanced doubling stages.

The seed at 1143.5 nm is an external cavity diode laser which is locked to a reference cavity (Finesse $\approx 10 \cdot 10^3$), made of ultra-low expansion glass. The UV light is amplitude controlled by an acousto optical modulator (AOM), which is also used for intensity stabilisation. For short pulses (as in Section 5) we use a sample and hold technique. The beam is focused onto the atoms with a waist of 20 $\mu$m with circular polarisation. The maximum power used is 38 mW for a Rabi frequency of $2\pi \cdot 400$ kHz.

## B Experimental sequence

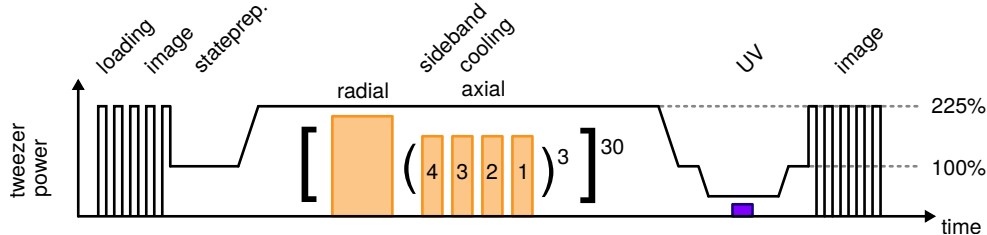

Figure 6: Experimental sequence. The chopping and ramps of the tweezer light (black lines), the Raman cooling light (yellow) and the UV pulse (purple) are shown. The numbers for the axial Raman sideband cooling indicate the $n^{th}$ sideband used for cooling, the brackets indicate the repetition of cooling cycles.

In Fig. 6 the experimental sequence is depicted. For loading and imaging single atoms, the tweezer light and D2 molasses light are chopped out of phase at 1.4 MHz [58, 60]. The tweezer light is stabilised to the average power during chopping, referred to as 100 % power in the main text. Also the state preparation in the $F = 2, m_F = 2$ state on the D1 line is performed chopped. For Raman sideband cooling and spectroscopy the trap is ramped to 225 % of the initial power (not chopped), which increases the trapping frequencies by a factor of 1.5. Raman sideband cooling consist of 30 cooling cycles. For each cycle we cool the radial axes for 2 ms with ten chirps of 200 $\mu$s over 120 kHz to cover the inhomogeneities between the different traps. Then we apply three sub-cycles of axial cooling, each cooling for 200 $\mu$s on the 4$^{th}$ to 1$^{st}$ axial sideband [55]. The whole cooling takes 150 ms, including switching times. The detuned optical repumpers for the $F$ and $m_F$ states are on during the whole cooling sequence.

## C Sideband spectroscopy

To quantify the cooling performance, we perform sideband spectroscopy of the motional sidebands. After Raman sideband cooling we apply a spectroscopy pulse to transfer the atoms from the $F = 2, m_F = 2$ to the $F = 1, m_F = 1$ state. The spectroscopy is performed in the same conditions as the cooling with the tweezers at 225 % of the initial power. Afterwards the traps are ramped to 20 % power and we heat out atoms in $F = 2, m_F = +2$ with resonant D2 light on the $F = 2, m_F = 2$ to $F' = 3, m_{F'} = +3$ cycling transition to only image the $F = 1$ population. For the radial spectroscopy we use a approximate $\pi/2$ pulse on the carrier while for the axial spectroscopy we apply a approximate $\pi$ pulse on the carrier. Both pulses are less than $\pi/2$ pulses on the respective sidebands. The results for radial and axial spectroscopy are shown in Fig. 7. From the sideband asymmetry we calculate the mean vibrational quantum number $\bar{n}$ by $\frac{\bar{n}}{\bar{n}+1} = \frac{I_{blue}}{I_{red}}$ with the strength of the blue and red sideband $I_{blue/red}$ [49]. The radial axes are cooled to $\langle \bar{n}_{rad} \rangle = 0.225 \pm 0.217$, where $\langle \cdot \rangle$ indicates the tweezer averaging which dominates

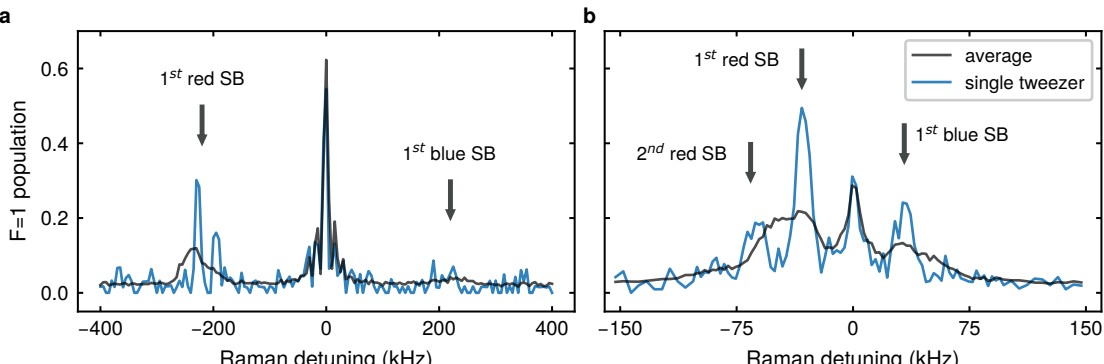

Figure 7: **a.** Radial spectroscopy of a single tweezer of the array (blue) and the averaged signal over all tweezers (black). The two peaks indicate a slightly elliptic tweezer with different radial trap frequencies. **b.** Axial spectroscopy for a single tweezer (blue) and averaged over all tweezers (black). For individual traps higher sidebands are also resolved.

the standard deviation. The large uncertainty is explained by the long-tailed distribution of the $\bar{n}_{rad}$ with a median of 0.142. Due to the lower trapping frequency in the axial direction we initially start outside the Lamb Dicke regime [55]. However, after cooling a clear asymmetry in the red and blue sideband is visible. The calculated mean vibrational quantum number is $\langle \bar{n}_{ax} \rangle = 1.04 \pm 1.05$ (median of 0.742). The best cooling for a single tweezer is $\bar{n}_{rad} = 0.13$ and $\bar{n}_{ax} = 0.23$, which gives a ground state probability of 69 %. Note that the cooling is optimised for best performance of the whole array.

Notably, we can also cool with comparable cooling performance with only two Raman beams. In this configuration we apply the same sequence for cooling as before, but only use the beams with projection on all trap axes (R2 + R3) for all cooling cycles (both radial and axial cooling).

## D Gravity and temperature calculation

To extract the temperature from the ramp-down measurement, presented in Fig. 3, we follow the analysis from Tuchendler et al. [63]: We first calculate the trap depth, taking into account gravity in the axial direction. At a factor $2 \cdot 10^{-4}$ of the initial power $P_0$ (with $V_0(P_0) = 1\,\mathrm{mK}$) gravity opens the trap and the atoms are not confined in the axial direction anymore. In the second step we calculate the minimum trap depth $V_{0,\mathrm{esc}}$ at which the atoms are lost depending on their initial energy $E_i$. For this we numerically solve the constant action equation $S(E_i, V_{0,i}) = S(V_{0,\mathrm{esc}}, V_{0,\mathrm{esc}})$. The action $S$ is defined as $S = \int_0^{x_{\max}} \sqrt{2m[E - V_0(x)]}dx$ with the Energy $E$ and axial potential $V(x)$. $x_{max}$ is the point where the atom has no kinetic energy. Finally, the survival probability $P_{\mathrm{surv}}(E) = \int_0^E f(E')dE'$ is extracted from a fit, with the Maxwell Boltzmann distribution $f(E)$. The curve shown in Fig. 3 is the extracted survival probability for the average of all tweezers as function of the optical power.

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
