# Peer review of "Raman Sideband Cooling in Optical Tweezer Arrays for Rydberg Dressing"

_SciPost Physics, doi:SciPost Phys. 10, 052 (2021)_

## Round 1 · Referee Report · Anonymous (Referee 1) · 2020-11-12

Strengths

1 - First demonstration of potassium atoms loaded into optical tweezers

2 - Demonstration of how Raman sideband cooling and adiabatic cooling can overcome the problem of inhomogeneous light shifts between the different tweezers

3 - The paper is clearly written.

Weaknesses

No major weaknesses to report

Report

The manuscript "Raman Sideband Cooling in Optical Tweezer Arrays for Rydberg Dressing" details the implementation of Raman sideband cooling for potassium atoms in tweezer arrays. This is the first experimental demonstration of potassium atoms in optical tweezers. The implementation of Raman sideband cooling, along with adiabatic ramping, has allowed for the problems associated with inhomogeneous tweezer arrays to be overcome.

I would recommend this for publication after responding to the few comments below. These include a few details missing that a reader could find useful

Requested changes

1 - Page 1 - Discussion of weak dressing regime. I think a reference could be useful here. Reference [45], for example, has a more in depth discussion on these equations
2 - Page 1 - The central hole in the objective lens is an interesting feature of the setup. Has such an objective lens been implemented before? If so a citation may be appropriate and if not I would ask if there are any complications that this may add to the creation of the tweezer arrays (unfocused part of beam?)
3 - Page 2 - Figure 1 caption. Scientific notation would be better to avoid confusion when using 20.000 (different meaning between German and English languages)
4 - Page 2 - "All Raman beams have linear polarization…", From the description it is a bit unclear to me what the polarizations of the beams are. What do you mean by in plane (which plane)? My assumption was the x-y plane, but the quantization axis is also in this plane. An alternative suggestion would be to show the polarization vectors in Figure 1(a)
5 - Page 4 - "...from a normal distributions..." -> "...from normal distributions..."
6 - Page 5 - Figure 4(a). Why are there only 2 simulation lines and 3 data sets?
7 - Page 6 - finesse 10.000 same comment as above about using scientific notation to avoid confusion
8 - Page 6/7 - the definition of the blue and red sidebands in Figure 7 and the equation from reference [49], seem to be in disagreement. Cooling would imply requiring $\bar{n}\rightarrow0$ meaning the ratio of $\bar{n}/(\bar{n}+1)\rightarrow0$. The equation implies that a small red sideband corresponds to cooling which is the opposite of that shown in Figure 7

---

## Round 1 · Referee Report · Anonymous (Referee 2) · 2020-11-24

Strengths

1 - timely topic of producing and controlling single-atom tweezer arrays 2- novel system using K atoms 3 - carefully conducted experiments and detailed analysis of the presented data 4 - points the way towards interesting future applications of this K-tweezer setup 5 - correct choice of journal and well-adapted content to a specific audience

Weaknesses

no specific weaknesses, the article is very well done

Report

In this work, the authors present a careful study of their novel K-atom tweezer array experiment. These single-atom tweezer experiments currently are of huge interest in the context of quantum simulation and information. While immense progress is made in a variety of setups, complex technical and fundamental challenges remain, some of which this new setup tries to solve by using potassium atoms. This enables various new solutions which the authors discuss in the manuscript. The current manuscript is a detailed study of the initial steps required to perform Rydberg quantum simulations in this setup and will be of great interest to other existing and upcoming experiments. I think the manuscript is ideally suited for this journal and I highly recommend its publicationin SciPost.

Requested changes

I have no specific comments or requests for changes. The paper is very well written and easy to follow. It is adressed at an expert audience which will find the details of the presentation very useful.

---

## Round 2 · Author Response

Dear Editors, Dear Referees,
Thank you for the correspondence and the reports on our manuscript. Below we detail the changes made in response to the referee reports.
Sincerely yours,
The Authors
Thank you for the correspondence and the reports on our manuscript. Below we detail the changes made in response to the referee reports.
Sincerely yours,
The Authors

---

## Round 2 · List of Changes

- We added references [42, 44, 45] where the weak dressing equations are introduced.
- We added a more detailed discussion about the hole in the objective and its effects on imaging and tweezer generation. This is indeed a novelty introduced by our setup.
- We now use scientific notation for the numbers in the figure captions.
- We clarified the description of the Raman beam polarizations, also adding arrows indicating the polarization in fig 1.
- We added the missing theory prediction in fig. 4a
- The ratio of the red and blue sideband strengths was inverted before. This is now corrected.

---

## Editorial Decision

published